# Clinical Spectrum Associated with Wolfram Syndrome Type 1 and Type 2: A Review on Genotype–Phenotype Correlations

**DOI:** 10.3390/ijerph18094796

**Published:** 2021-04-30

**Authors:** Maurizio Delvecchio, Matteo Iacoviello, Antonino Pantaleo, Nicoletta Resta

**Affiliations:** 1Metabolic Diseases, Clinical Genetics and Diabetology Unit, Giovanni XXIII Children’s Hospital, 70126 Bari, Italy; 2Department of Biomedical Sciences and Human Oncology (DIMO), Division of Medical Genetics, University of Bari “Aldo Moro”, 70124 Bari, Italy; m.iacoviello3@gmail.com (M.I.); ninotp90@live.it (A.P.); nicoletta.resta@uniba.it (N.R.)

**Keywords:** Wolfram syndrome, wolframin, ERIS, *CISD2*, molecular genetics, genotype-phenotype correlation, diabetes mellitus, optic atrophy, sensorineural hearing loss, diabetes insipidus

## Abstract

Wolfram syndrome is a rare neurodegenerative disorder that is typically characterized by diabetes mellitus and optic atrophy. Other common features are diabetes insipidus and hearing loss, but additional less-frequent findings may also be present. The phenotype spectrum is quite wide, and penetrance may be incomplete. The syndrome is progressive, and thus, the clinical picture may change during follow-up. Currently, two different subtypes of this syndrome have been described, and they are associated with two different disease-genes, *wolframin* (*WFS1*) and *CISD2*. These genes encode a transmembrane protein and an endoplasmic reticulum intermembrane protein, respectively. These genes are detected in different organs and account for the pleiotropic features of this syndrome. In this review, we describe the phenotypes of both syndromes and discuss the most pertinent literature about the genotype–phenotype correlation. The clinical presentation of Wolfram syndrome type 1 suggests that the pathogenic variant does not predict the phenotype. There are few papers on Wolfram syndrome type 2 and, thus, predicting the phenotype on the basis of genotype is not yet supported. We also discuss the most pertinent approach to gene analysis.

## 1. Introduction

Wolfram syndrome (WS) is a rare neurodegenerative disorder that is sometimes referred to as Diabetes Insipidus, Diabetes Mellitus, Optic Atrophy, and Deafness (DIDMOAD), and it was described for the first time by Wolfram and Wagener in 1938 [1]. They described four siblings who were born to consanguineous parents with diabetes mellitus (DM) and optic atrophy, which are key elements that are used to diagnose this syndrome. In 1966, Rose et al. [2] reviewed the literature and suggested that this syndrome could be due to homozygous mutation in the disease gene. Furthermore, in consideration of the clinical heterogeneity and wide clinical spectrum involving different organs, they hypothesized that one or more additional genes could play a role in modulating the phenotype. In 1994, more than 50 years after the syndrome was first described, Polymeropoulos et al. [3] described the WS locus on the short arm of chromosome 4. Four years later, Inoue et al. [4] described the wolframin gene (*WFS1*; 606201) as the WS disease gene. They reported that *WFS1* encodes a transmembrane protein in beta cells and neurons, which explains the pleiotropic features of the syndrome. In 2000, El-Shanti et al. [5] provided evidence that there are two different subtypes of WS. They studied Jordanian siblings who were born to consanguineous parents featuring the typical WS phenotype and gastrointestinal symptoms, and they provided conclusive evidence of the existence of a second autosomal recessive form of WS, called WS type 2 (WFS2; 604928). A linkage study later showed that a second locus could be localized in 4q22-q24. In 2007, Amr et al. [6] showed that the WFS2 disease gene is the *CISD2* gene (611507.0001), which encodes the endoplasmic reticulum (ER) intermembrane small (ERIS) protein that is located in the ER.

In this review, we aimed to provide an overview the pertinent literature on the genetics of WS. In the first part of the manuscript, we present the clinical history of WS types 1 and 2 and the differences between them. We highlight the key points in a differential diagnosis and that support the choice of which gene should be analyzed. In the second part of the manuscript, we discuss the genotype–phenotype correlation for each gene.

## 2. Methods

A systematic literature review of the PubMed database up to 31 December 2020 was performed independently by two of the authors (M.I. and A.P.) to identify relevant papers. The search terms were “Wolfram syndrome” AND “CISD2” or “wolframin” or “ERIS”. Only papers on clinical and genetic findings about WS were considered for inclusion in this review. The search included clinical case reports, clinical case series, observational studies, and reviews. Potentially relevant papers were initially evaluated by checking the title and abstract, and all eligible studies were retrieved. For the purpose of this review, all papers that clearly stated that the WS diagnosis was genetically confirmed were considered to be eligible. Any other clinical description without genetic data or a Wolfram-like syndrome description was not considered. Additional papers were identified using a manual search of the references from the retrieved articles. Non-English manuscripts were not included in the search and review. This article is based on previously conducted studies, and it does not contain any studies with human participants or animals that were performed by any of the authors. Figure 1 shows the article selection process.

## 3. Epidemiology

WS is very rare. The first epidemiological study was a nationwide study that was conducted in the UK in 1995, and it showed a prevalence of one in 770,000 subjects [7]. This study showed results that were similar to data from Japan, which indicated a prevalence of one in 710,000 [8]. The prevalence is reported to be strikingly higher in Lebanon, where Zalloua et al. [9] found 22 patients with a genetically confirmed diagnosis of WFS1 out of 399 (5.5%) with juvenile onset diabetes. Other papers reported a lower prevalence of WS (<0.5%) in patients with DM from Italy [10,11] and China [12], and this prevalence was even lower (0.04%) in a large international multi-center pediatric diabetes registry [13]. No epidemiological data are available about WFS2 because patients have not been diagnosed in all countries. Only one paper distinguished between the two types, and four patients with genetically confirmed WS were reported, three with WS type 1 and one with WS type 2 [11].

## 4. Clinical Findings and Treatment

WS is a progressive neurodegenerative disorder, which should always be suspected in patients with insulin-dependent DM and optic atrophy. DM often occurs before 10 years of age. Diabetic ketoacidosis is rare, the insulin requirement is low, and the clinical course is not progressive, and it is also milder than type 1 DM [14]. Microvascular complications are uncommon, and they are likely related to residual insulin secretion. Unfortunately, hypoglycemia episodes may be frequent due to neurologic dysfunctions, which may lead to hypoglycemia unawareness. The treatment is based on the basal-bolus insulin regimen and the metabolic control is effective even on low doses [15].

Optic atrophy is progressive, and it is usually diagnosed before 15 years of age. It is characterized by a progressive decrease in visual acuity with a color vision defect, which leads to blindness. No treatment is currently available to stop the progression of eye involvement. Less frequent findings may include cataract, nystagmus, and pigmentary retinopathy [15].

Besides these two key elements for the diagnosis of WS, several other possible clinical findings that involve different organs may occur. The most frequent occurrence is sensorineural hearing loss, which is estimated to involve about two-thirds of these patients. The clinical spectrum may range from congenital deafness to mild impairment, which is sometimes progressive as a consequence of the central nervous system degenerative process [7]. It is usually diagnosed in the second decade of life. The audiogram typically shows a downward sloping progressive pattern of hearing loss [16]. Regular monitoring is suggested for appropriate treatment, and hearing aids and cochlear implants may be therapeutic tools for these patients [17].

DM is not the only endocrine disease in WS. Diabetes insipidus is frequent, and it occurs mostly in the second decade of life. It is characterized by the loss of ability to concentrate urine, leading to low osmolality urine with polyuria. It can be also partial, making this diagnosis more difficult, and thus, it is often delayed. Most of the patients respond well to treatment with desmopressin [18]. Male patients may present hypogonadism more frequently than female patients, secondary to hypothalamus–pituitary axis impairment or gonadal failure. Hypothyroidism and growth retardation have also been reported, and some pregnant patients have been described [15].

Neurologic abnormalities occur later, usually in the third and fourth decade of life in about 60% of the patients. A more detailed evaluation may show subclinical neurological abnormalities even in earlier stages of the disease (late puberty), and thus, the mean age at onset of these abnormalities is currently considered to be more precocious than in the past. They are progressive, leading to general brain atrophy, which is more prominent in the cerebellum, pons, and medulla, and there is brain stem and cranial nerve involvement [7,19,20]. There is no evident correlation between neurological imaging and clinical findings [21]. Other uncommon findings may be truncal or gait ataxia, central apnea, dementia, and intellectual disability. A significant increase in suicidal behavior and psychiatric illness has been reported [22]. Neurological abnormalities may also involve the urinary tract, causing neurogenic bladder with hydroureter, urinary incontinence, and recurrent infections. Incomplete bladder emptying or complete bladder atony may be detected by a urodynamic examination, which is required in patients with these symptoms.

De Heredia et al., analyzed clinical and genetic data from 412 patients with WS who were reported in the literature in the previous 15 years [23]. They showed that 98.2% of the patients had DM, 82.1% had optic atrophy, 48.2% had deafness, and 37.8% had diabetes insipidus. Urological manifestations and neurological symptoms were described in 19.4% and 17.1% of patients, respectively. However, the phenotype depends on the age of the patient, and thus, less frequent clinical findings may become more frequent in older patients. Death occurred at a median age of 30 years, with two frequency peaks around 24 and 45 years.

Life expectancy is shortened. About 65% of WFS1 patients die before 30–40 years and the average age of death is 30 (range 25–49) years [24].

The clinical picture that is reported above is the typical WS phenotype that is described in the literature. Over the past two decades, upper intestinal ulcers and defective platelet aggregation have been reported in some patients. These clinical findings are absent in WFS1 and are pathognomonic of WFS2 [25,26] and, thus, they represent important points for a differential diagnosis, which leads to analysis of the *CISD2* rather than the *WFS1* sequence. They have been observed in up to 90% of patients with WFS2 [27].

As in WFS1, optic atrophy is progressive and associated with the loss of ganglion cells, even if the eye impairment in WFS2 is milder and even if there is less progression [5]. Mozzillo et al., discussed eye involvement and provided data suggesting that involvement of the optic nerve is compatible with a diagnosis of optic neuropathy rather than that of optic atrophy [26].

Differences in the phenotype between WFS1 and WFS2 are related to a difference in tissue expression of *WFS1* and *CISD2.* Table 1 summarizes the typical clinical findings in both types.

### Treatment Perspective

Each disorder that is associated with WS can be properly treated, but the main goal of a specific treatment for WS would be to stop disease progression in all the involved tissues [29]. Possible specific drugs should aim to prevent cellular aging and the degenerative process, by the maintenance of ER, calcium homeostasis, protein folding, and redox regulation [29,30,31]. Current ongoing projects in regenerative medicine and gene therapy are great challenges for the treatment of WS and all neurodegenerative disorders [28], but the clinical application seems to remain unchanged. To date, there are no pharmacological therapies for WS.

The most suitable treatment strategy seems to be based on chemical chaperones, which play a role in protein folding in the ER. Two chemical chaperones, 4-phenylbutyricacid (PBA) and tauroursodeoxycholic acid (TUDCA), are under investigation. They preserve beta cell function by reducing stress and cell death. Furthermore, they slow down the neurodegeneration process [32]. Another approach is based on the prevention of calcium-mediated ER stress, and in turn, prevention of cell death, by modulating intracellular calcium levels. Dantrolene, which is used for some neurological disorders, suppresses the efflux of calcium from the ER to the cytosol, which preserves the integrity of beta and neural cells [33]. A similar effect could be exerted by drugs that bind to the sarco/ER Ca^2+^-ATPase, which is a substrate of Wolframin, or that target the calcium channel receptor that is activated by inositol triphosphate [34]. Finally, valproate acid is also being investigated as a novel drug treatment for neurodegeneration and diabetes in WS. Valproate acid is neuroprotective, and recently, a phase 2 clinical trial in patients with WS has been started.

Besides these experimental trials, different hypoglycemic agents, which are already licensed for DM, have been shown to improve glycemic control in these patients with limited and controversial effects on the generation process [28]. However, their use seems to be effective only on blood glucose homeostasis and not on the pathogenesis of WS.

## 5. Molecular Genetics

*WFS1* is a nuclear gene that is composed of eight exons (a total of 33.4 kb), and it is located on chromosome 4 (4p16). This gene encodes for Wolframin, a protein that is embedded in the ER, and it is composed of nine transmembrane segments and large hydrophilic parts at each end [4,35]. Wolframin is highly expressed in pancreatic beta cells, heart, lung, placenta [18], and brain tissue [36].

Wolframin serves as an ER calcium channel, suggesting that this protein may play a role in ER homeostasis [37]. The ER is required for the folding and the secretion of newly synthesized secretory proteins such as proinsulin. ER stress, which is defined as an imbalance between the actual folding capacity of the ER and the folding demand placed on this organelle, causes activation of a cell death pathway through the unfolded protein response (UPR) [37]. The authors found that Wolframin protects beta cells against ER stress and, conversely, that ER chronic stress is caused by loss of function of WFS1. They also report that wolframin mRNA is induced by ER stress to maintain ER homeostasis and to prevent the death of pancreatic beta cells. *CISD2* maps to chromosome 4 (4q22-q24), contains three exons (a total of 23 kb), codes for ERIS, which is embedded in mitochondria-associated ER membranes, and contains a transmembrane domain at the N-terminal and a single CDGSH domain at the C-terminal [38,39]. ERIS has a role in maintaining both the structural integrity and the functional cross-talk between the ER and mitochondria, acquiring a pivotal role in the regulation of glucose homeostasis and insulin sensitivity [40,41]. This protein is expressed in the pancreas, the brain, and other tissues [42,43].

Both Wolframin and ERIS seem to share an overlapping function and have important roles in regulating intracellular calcium homeostasis and the ER stress response [18,44,45].

### 5.1. Wolfram Syndrome Type 1 Genotype–Phenotype Correlations

*WFS1* was identified as a disease gene in 1998, and since then, more than 170 different mutations have been identified [23]. Most mutations are situated on exon 8 and are inactivating mutations such as nonsense or frameshift mutations [46]. One of the most relevant studies about WS was performed by de Heredia et al. [23]. They reviewed clinical and genetic data from 412 patients from 49 studies and found 178 mutations in *WFS1*, which mostly involves the Wolframin N-end, transmembrane domains, and the last 100 amino acids. These mutations have been classified based on their effect on *WFS1* expression [23] (see Table 2 and Table 3). Many other studies have tried to establish a genotype–phenotype correlation, which is a difficult task due to the molecular complexity of *WFS1*, the different clinical characteristics, and the small size of patient cohorts [46]. Basically, the clinical history may be different even among patients within the same pedigree, and the type or location of the pathogenic variant does not predict the phenotype.

WS1 patients can be homozygous or compound heterozygotes. Compound heterozygous patients have a higher risk of psychiatric disorders, DM, and hearing loss [23]. The inheritance pattern of *WFS1* mutations together with the type of mutation that is inherited can affect both the onset and the severity of the main clinical features of WS [47]. Some studies have found that patients who are homozygous or compound heterozygotes for two inactivating mutations are characterized by an earlier onset of DM [14,24] and optic atrophy [14]. Furthermore, compound heterozygosity for missense mutations may lead to a mild phenotype [23,48].

Patients who have complete loss-of-function mutations seem to develop DM at an earlier age than those who have partial loss-of-function mutations [8]. Moreover, it seems that these mutations are predictive of isolated diabetes, isolated deafness, or isolated congenital cataracts without the development of the full syndrome [49]. *WFS1* mutations are also responsible for “uncommon” WFS1 phenotypes such as autosomal dominant low-frequency sensorineural hearing loss or an association between optic atrophy and deafness [24,47,50].

There is ample literature about unusual WFS1 presentations. For example, Berry et al. [51] reported a family with isolated congenital cataracts that were associated with a missense mutation, which was inherited in an autosomal dominant manner. Bonnycastle et al. [52] reported another family with autosomal dominant, isolated, adult-onset diabetes secondary to another missense mutation. Elli et al. [53] found a complex structural rearrangement in a patient with neonatal diabetes insipidus, optic tract hypoplasia, psychomotor retardation, and central hypothyroidism. They found a new chromosomal abnormality in the absence of any *WFS1* and *CISD2* mutations that were previously reported. They suggested the possibility of uniparental heterodisomy (UPD) after observing non-Mendelian transmission of some polymorphisms in the promoter region. The authors tested this hypothesis by performing an analysis of genetic markers along the WFS1 region, and they showed evidence of a segmental paternal UPD, involving at least part of the promoter and the first exon. Chaussenot et al. [47] described a group of patients with an unusual presentation of WS. One patient was homozygous for a missense mutation and presented with optic atrophy (at 10 years of age), cerebellar ataxia, and neurogenic bladder (at 27 years of age), but no DM. Another patient was heterozygous for a single mutation and developed DM and deafness at 1 year of age, followed by glaucoma, bilateral cataracts, cerebellar ataxia, areflexia, short stature, hypothyroidism, and hypogonadism. Finally, 12 patients with late-onset WS (LOWFS) developed DM and optic atrophy at 15 years of age or later and they carried at least one *WFS1* mutation. Although it is not a main feature of WFS1, other studies have found heterozygous *WFS1* mutations that were associated with cataracts [54,55]. Figure 2 displays the genetic variant distribution in *WFS1*.

Papadimitriou et al. [56] reported a case of WFS1 that was caused by maternal UPD of chromosome 4 with a homozygous mutation in *WFS1*. The patient was a 10-year-old girl who developed an usual WFS1 presentation. She was diagnosed with type 1 DM (since 6 years of age), high-frequency sensorineural hearing loss, and a reduction in visual acuity. Her parents and her brother were all healthy. The mutation that was found in the patient was heterozygous in her mother but neither her father nor her brother were carriers. The authors then performed a combination of a single nucleotide polymorphism (SNP) array for genotyping and a microsatellite analysis to determine the origin of the second allele. The analysis indicated isodisomy due to uniparental segregation of the maternal alleles that may result from either meiosis II or post-zygotic duplication. Another issue related to the understanding of WFS1 genetics is the reduced penetrance in the dominant form because not all carriers of mutations develop the disease [50,57]. Riachi et al. [58] hypothesized about the involvement of a mechanism such as nonsense-mediated decay, methylation signatures, or post-transcriptional regulations in the modulation of penetrance.

### 5.2. Wolfram Syndrome Type 2 Genotype–Phenotype Correlations

*CISD2* mutations are involved in WFS2, which is characterized by DM, optic atrophy, and deafness with an absence of diabetes insipidus and psychiatric disorders [5,6,25,39,59,60]. The locus was mapped for the first time in three large consanguineous Jordanian families [5]. Individuals with WFS2 show additional features that have not been described in patients with WFS1, such as defective platelet aggregation with collagen, bleeding tendency, and peptic ulcer disease [6,26,61]. Platelet aggregation is calcium mediated and the Human Protein Reference Database (accession number 17413) shows the presence of *CISD2* transcripts in platelets. This may explain, at least in part, the bleeding phenotype [26].

The first Caucasian patient who was affected by WFS2 was described by Mozzillo et al. [26]. She was a 17-year-old patient who had DM, optic neuropathy, intestinal ulcers, sensorineural hearing loss, and defective platelet aggregation responses to adenosine diphosphate (ADP). A novel homozygous intragenic deletion of *CISD2* was found in the proband, but her parents and brother were heterozygous for the same mutation and were apparently healthy except for subclinical defective platelet aggregation.

Rondinelli et al. [61] reported a novel *CISD2* mutation that caused WFS2 in two Italian sisters. They found a homozygous substitution in both of them, while the parents carried the same mutation in a heterozygous state. Severe duodenal ulcers were the first symptom, and the two patients presented DM, optic atrophy, hearing loss limited to high frequencies, and polyuria. Other features included hypogonadism, rheumatologic diseases, and hypogammaglobulinemia. Figure 3 displays the genetic variant distribution in *CISD2*.

Recently, Pourreza et al. [60] discovered another novel pathogenic variant of *CISD2*, which is a homozygous substitution in a patient who was born of a consanguineous marriage to parents with WFS. The patient developed DM, optic atrophy, polyuria, and upper intestinal ulcers. The results presented thus far suggest a possible genotype-phenotype association that is peculiar to WFS2. It seems that all patients with *CISD2* mutation develop peptic ulcer disease and bleeding tendency.

Conversely, Rouzier et al. [39] reported a Moroccan patient with an overlapping phenotype, which suggests that WFS1 and WFS2 form a clinical spectrum with genetic heterogeneity. They have identified a novel homozygous *CISD2* mutation in a patient with “classical” WFS1 phenotype, which is childhood-onset DM and optic atrophy without peptic ulcers or defective platelet aggregation, as reported previously in *CISD2* mutation carriers. Rouzier et al.’s [39] results suggested screening for *CISD2* in patients who manifest the defining features of WS. In conclusion, further studies are necessary to better define the clinical aspects of WF1 and WF2.

## 6. Genetic Analysis of Wolfram Syndrome

According to the literature that was examined in this review, WS is caused by *WFS1* and *CISD2* mutations. Therefore, if WFS1 is suspected, it is appropriate to proceed to the sequencing of all eight of the exons and their flanking intronic regions. Similarly, if WFS2 is suspected, it is useful to sequence the three *CISD2* exons and their flanking intronic regions.

Additionally, WS could be caused by point mutations and, in some cases, by the deletion of an entire exon. Thus, it is recommended to perform an exhaustive molecular analysis that is aimed to identify any type of genetic alteration including quantitative imbalance such as “copy number variation” (CNV) as well as the possible presence of UPD.

## 7. Conclusions

WS is a rare and severe neurodegenerative disorder that involves different organs. To consider the pleiotropic manifestations, a multidisciplinary, team approach to the different clinical problems should be used for these patients. This diagnosis should be considered in all the patients with DM and optic atrophy and with an overall absence of type 1 DM autoantibodies. We recommend that this diagnosis should also be suspected in patients with non-autoimmune DM who present less frequent clinical findings that are suggestive of WS, such as sensorineural hearing loss or bowel disease. We recommend particular attention is given to patients with likely WS and bowel disorders or a wild-type wolframin gene sequence because WFS2 is not as well-known and, thus, it is investigated less frequently. Furthermore, *CISD2* gene sequencing is not performed in all the laboratories, and thus, WFS2 could be underdiagnosed.

From a basic science point of view, WS is an interesting model to investigate drugs and molecules that are involved with ER homeostasis and cellular senescence. Inflammatory pathologies, DM, atherosclerosis, neurodegenerative diseases, and even cancer are related to ER dysfunction. More insights into these mechanisms could be interesting for translational research.

An early diagnosis allows proper genetic counseling and proper follow-up to occur, which prompts clinicians to search for possible associated disorders. Unfortunately, the genotype does not predict the phenotype, and determining a prognosis for these patients is difficult.

Finally, we would like to comment about the possibility of diagnosing WS at pre-clinical or paucisymptomatic stage. Increasing knowledge about the genetic mechanisms of monogenic DM allows exome sequencing for a molecular diagnosis, which allows the diagnosis of many monogenic subtypes including WS. Besides the great advantages of this technique, more attention should be paid in the diagnosis of an untreatable neurodegenerative disorder, such as WS. What could be the backlash of a neurodegenerative disorder in a patient with DM? What is the advantage of diagnosing WS in a patient with non-autoimmune diabetes without eye involvement? Shall we tell a patient: “You have diabetes but it is very likely you will become blind; we do not know when, and we cannot do anything to prevent it”? This review highlights that the genetic variant will not strictly predict the clinical findings and, thus, we are not able to provide a reliable prognosis for patients with diabetes but without any other clinical manifestation.

## Figures and Tables

**Figure 1 ijerph-18-04796-f001:**
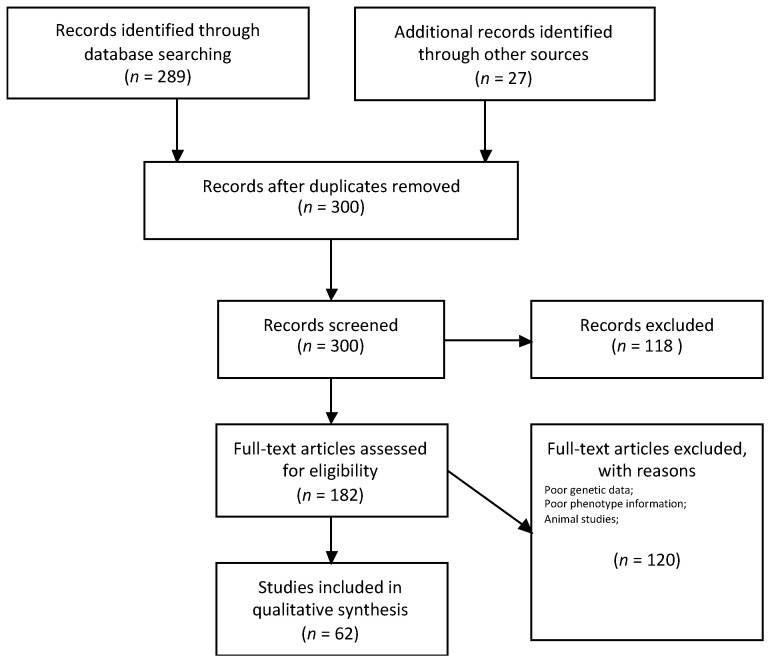
Flow diagram for article selection.

**Figure 2 ijerph-18-04796-f002:**
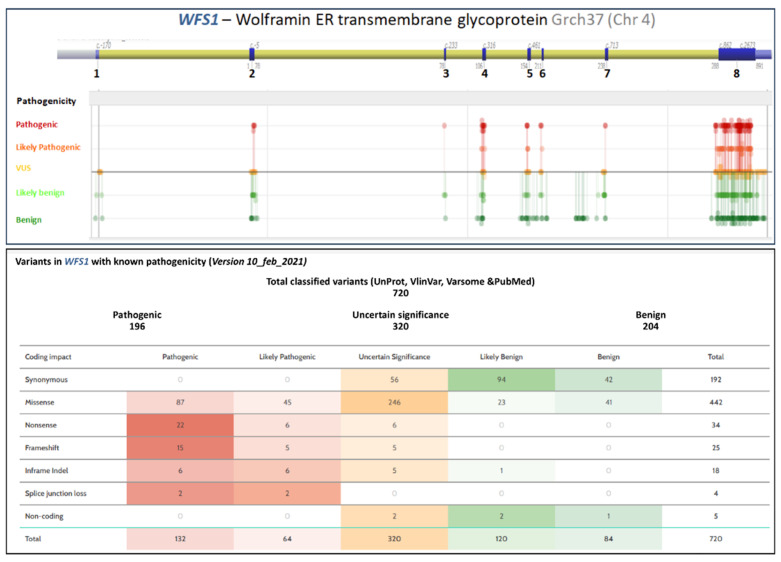
Genetic variant distribution in *WFS1.* Adapted from Alamut Visual version 2.15 (SOPHiA GENETICS, Lausanne, Switzerland) and from *VarSome: The Human Genomic Variant Search Engine*. Christos Kopanos, Vasilis Tsiolkas, Alexandros Kouris, Charles E. Chapple, Monica Albarca Aguilera, Richard Meyer, and Andreas Massouras. *Oxford Bioinformatics*, bty897, 30 October 2018. doi:10.1093/bioinformatics/bty897.

**Figure 3 ijerph-18-04796-f003:**
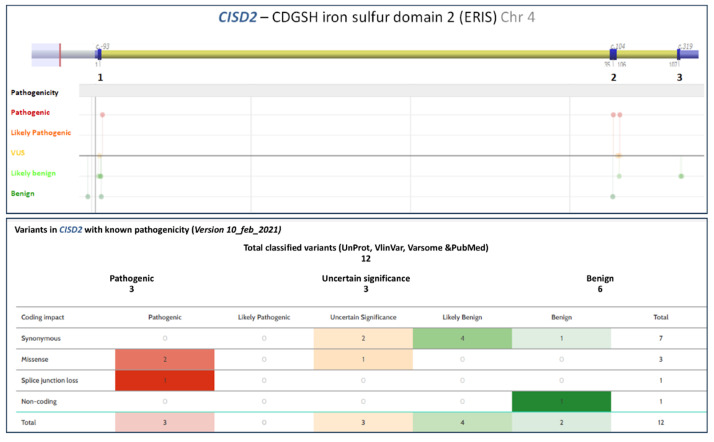
Genetic variant distribution in *CISD2*. Adapted from Alamut Visual version 2.15 (SOPHiA GENETICS, Lausanne, Switzerland) and from *VarSome: The Human Genomic Variant Search Engine*. Christos Kopanos, Vasilis Tsiolkas, Alexandros Kouris, Charles E. Chapple, Monica Albarca Aguilera, Richard Meyer, and Andreas Massouras. *Oxford Bioinformatics*, bty897, 30 October 2018. doi:10.1093/bioinformatics/bty897.

**Table 1 ijerph-18-04796-t001:** Clinical features of Wolfram syndrome type 1 and type 2 and age at onset. Major clinical findings are reported in order of age at onset in the clinical history. Adapted from Pallotta et al. [28].

Major Clinical Findings	Other Clinical Findings
Diabetes mellitus (a) Age at diagnosis: 6–10 years	Urinary tract problems and renal dysfunction (neurogenic bladder, bladder incontinence, urinary tract infection) Age of diagnosis: second decade of life
Optic atrophy (a) Age at diagnosis: 10–15 years	Psychiatric symptoms (depression, psychosis, panic attacks, sleep abnormalities, mood swings)
Diabetes insipidus Age at diagnosis: 14–20 years	Neurological manifestation/autonomic dysfunction (central apnea, dysphagia, areflexia, epilepsy, decreased ability to taste and detect odors, headache, orthostatic hypotension, hypothermia, hyperpyrexia, gastroparesis, constipation)
Sensorineural hearing loss Age at diagnosis: 16–20 years	Endocrine disorders (hypogonadism, growth hormone deficiency, corticotropin deficiency, delayed menarche)
Ataxia Age at diagnosis: 15–25 years	Dominant disease with or without diabetes mellitus and recessive Wolfram-like disease without diabetes mellitus
Upper intestinal ulcers and platelet aggregation defect (b)	

(a) required for the diagnosis of Wolfram syndrome. It has been suggested that “optic neuropathy” could be more appropriate in Wolfram syndrome type 2. (b) absent in Wolfram syndrome type 1, typical of Wolfram syndrome type 2.

**Table 2 ijerph-18-04796-t002:** Classification of Wolfram Syndrome type 1 mutations [23]. Adapted from Rigoli et al. [46].

Groups	Localization	Type	Alterations
1	before exon 8	nonsense and frameshift	complete deletion
2	aa 1–670 aa 701–890	missense nonsense	complete degradation
3	after exon 8 and before aa700 after exon 8 aa 671–700	nonsense frameshift missense	defective or trun-cated protein

**Table 3 ijerph-18-04796-t003:** Functional alterations of Wolframin [23]. Adapted from Rigoli et al. [46].

Class	Functional Alterations
A1	Wolframin depletion due to *WFS1* mRNA degradation
A2	Wolframin depletion due to *WFS1* mRNA and protein degradation
A3	Wolframin depletion due to protein degradation
B	Reduced expression of defective Wolframin
C	Expression of defective Wolframin

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
