# Peer review of "Clinical Spectrum Associated with Wolfram Syndrome Type 1 and Type 2: A Review on Genotype–Phenotype Correlations"

_ijerph, 2021, doi:10.3390/ijerph18094796_

Round 1
Reviewer 1 Report
The review by Delvecchio M et al. summarizes clinical evidence and genetic bases about Wolfran Syndrome Type 1 and 2. I believe this is an important contribution but quite specific to the field.
Some concerns should be adressed by the auhors:
1) In Figure 1 questions marks appear over the text and must be eliminated.
2) Pag 3/12 L14-18 Please cite the adecuate refererences about clinical presentation of the syndrome
3) Pag 3/12 L22-26 Please cite the adecuate refererences about optic atrophy description
4) Pag 3/12 L39-42 please cite adecuate references.
5) Pag 5/12 L 44 The link does not work
6) Is it known how CISD2 or WFS2 would be involved in platelet agregation?
Reviewer 2 Report
Congratulations to the authors for the well-written review article on Wolfram syndrome. The article discusses the genotype-phenotype of Wolfram syndrome.
Have few minor suggestions
- Can you explain the clinical presentations of Diabetes Insipidus?
- Not sure about the importance of mentioning 'some pregnant patients have been described in the clinical finding and treatment section.
- Is there any data available about the life expectancy of the patients with Wolfram syndrome?
- How difficult is it to control the blood sugars in these patients?
- If possible provide reference to - overall absence of type 1 DM antibodies
